# SCALING LAWS FOR UNCERTAINTY IN DEEP LEARNING

## ABSTRACT

Scaling laws in deep learning describe the predictable relationship between a model's performance, usually measured by test loss, and some key design choices, such as dataset and model size. Inspired by these findings and fascinating phenomena emerging in the over-parameterized regime, we investigate a parallel direction: do similar scaling laws govern predictive uncertainties in deep learning? In identifiable parametric models, such scaling laws can be derived in a straightforward manner by treating model parameters in a Bayesian way. In this case, for example, we obtain $O(1/N)$ contraction rates for epistemic uncertainty with respect to dataset size $N$. However, in over-parameterized models, these guarantees do not hold, leading to largely unexplored behaviors.

In this work, we empirically show the existence of scaling laws associated with various measures of predictive uncertainty with respect to dataset and model size. Through experiments on vision and language tasks, we observe such scaling laws for in- and out-of-distribution predictive uncertainty estimated through popular approximate Bayesian inference and ensemble methods. Besides the elegance of scaling laws and the practical utility of extrapolating uncertainties to larger data or models, this work provides strong evidence to dispel recurring skepticism against Bayesian approaches: *"In many applications of deep learning we have so much data available: what do we need Bayes for?"* Our findings show that *"so much data"* is typically not enough to make epistemic uncertainty negligible.

## 1 INTRODUCTION

Deep learning has recently revealed empirical scaling laws: test performance tends to scale by a power-law $f(x) \propto x^{-\gamma}$ of the size of the data, model parameters or compute (Kaplan et al., 2020; Hoffmann et al., 2022). A related 'double descent' effect shows that increasing model capacity can surprisingly improve generalization (Belkin et al., 2019). These findings suggest that bigger models often perform better. A parallel question is: do similar scaling laws govern uncertainty in deep learning?

Bayesian deep learning provides a principled framework for quantifying uncertainty in neural networks by marginalizing weights, rather than

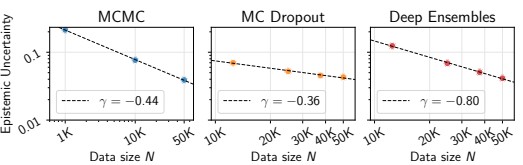

Figure 1: **Deep learning uncertainty is predictable, empirically**. ResNet-18 Epistemic Uncertainty scaling with the number of training data $N$ on `CIFAR-10`.

relying on point estimates, to obtain predictive uncertainties. (Neal, 1996; MacKay, 1995). Recent contributions show that, under rather simple conditions, inferring only a subset of model parameters can retain the capacity to represent any predictive distribution, at a significantly lower cost (Sharma et al., 2023)

The ability to produce well-calibrated predictive distributions and uncertainties is essential in decision-making and safety-critical applications, such as medical diagnosis and autonomous driving, where sound quantification of uncertainty can guide downstream choices (Papamarkou et al., 2024). However, exact Bayesian inference in large networks is intractable, which has motivated scalable approximations such as those based on the Laplace method (Ritter et al., 2018; Antoran et al., 2022), Variational Inference (VI) (Graves, 2011; Shen et al., 2024), Monte Carlo (MC) Dropout (Gal &

Ghahramani, 2016), along with (non-Bayesian) Deep Ensembles (Lakshminarayanan et al., 2017), which have proven effective and practical in obtaining sensible predictive uncertainties. Sampling-based methods, such as Stochastic Gradient Langevin Dynamics (SGLD) (Welling & Teh, 2011) and Stochastic Gradient Hamiltonian Monte Carlo (SGHMC) (Chen et al., 2014) provide scalable Markov chain Monte Carlo (MCMC) techniques for posterior inference over neural network weights. In the remainder of this paper, we will generally refer to Uncertainty Quantification (UQ) methods to indicate Bayesian and non-Bayesian approaches designed to obtain predictive uncertainties.

Despite recent criticism (Wimmer et al., 2023), current trends have focused on information-theoretic decomposition of predictive uncertainties into aleatoric and epistemic parts (Hüllermeier & Waegeman, 2021). In this work, we study how *predictive uncertainties* evolves in deep models as dataset size $N$ and model size $P$ grow, evaluating a range of UQ methods and architectures for both vision and language modalities. With stronger emphasis on scaling with respect to $N$, we systematically vary optimization settings, prior choices, stochasticity level, and inference techniques, demonstrating that different configurations exhibit a range of scaling behaviors across uncertainty metrics, but crucially all of them follow power-laws (see, e.g., Fig. 1).

While classic results show $O(1/N)$ posterior-variance contraction in identifiable parametric models (Cam, 1953), over-parameterized neural networks present challenges due to singular likelihoods, non-identifiability, and non-Gaussian posteriors. Although empirical scaling laws apply to test performance (Kaplan et al., 2020), a systematic understanding of how uncertainty scales with $N$, model size $P$, or computational budget is lacking. Both Bayesian and non-Bayesian UQ methods yield ensembles, with Epistemic Uncertainty (EU) reflecting prediction diversity. One of the interesting aspects of scaling laws for uncertainty is to identify when EU vanishes, indicating ensemble collapse.

Several theories characterize neural networks in the large-data limit. Singular Learning Theory (SLT) models deep networks as singular statistical manifolds, claiming that singularities of loss minima determine learning behavior, and that generalization follows a power-law $\mathcal{L} - \mathcal{L}_0 = \frac{\lambda}{n^\gamma}$ (Watanabe, 2009). In this expression, $\mathcal{L}$ indicates the test loss, $\mathcal{L}_0$ the theoretical minimal achievable loss (e.g., due to noise in the data), $\lambda$ a constant that depends on the model and $\gamma$ the Real Log Canonical Threshold (RLCT) or learning coefficient. This last quantity governs the rate at which the generalization loss approaches the irreducible loss as $n$ (number of training data points) grows. Other perspectives include the manifold dimensionality hypothesis, linking generalization to the intrinsic dimensionality of learned representations (Ansuini et al., 2019), and mechanistic interpretability efforts explaining simple scaling phenomena via emergent internal circuits (Nanda et al., 2023). We derive a formal connection of generalization and Total Uncertainty (TU) in the case of linear models. We believe that such a framework may help to explain the scaling behavior of uncertainty in deep models.

In summary, our contributions are as follows:

(i) *Empirical study.* We provide a comprehensive evaluation of predictive uncertainties using a variety of UQ methods across different architectures, modalities, and datasets. To the best of our knowledge, this is the first study to consider scaling laws associated with any form of uncertainty in deep learning.

(ii) *Scaling patterns.* We empirically demonstrate that predictive uncertainties evaluated on in- and out-of-distribution, follow power-law trends with dataset and model size. This allows us to extrapolate to large dataset sizes and to identify data regimes where UQ approaches remain relevant to characterize the diversity of the ensemble to a given numerical precision.

(iii) *Theoretical insights.* We derive a formal connection between generalization error in SLT and TU in linear models. This novel analysis provides an interesting lead to explain the scaling laws observed in the experiments for over-parameterized models.

## 2 BACKGROUND

### 2.1 UNCERTAINTY QUANTIFICATION

Predictive uncertainty refers to metrics associated with an ensemble of predictive distributions and it can be decomposed into Aleatoric Uncertainty (AU), arising from intrinsic data variability, and EU, reflecting uncertainty due to limited data or model knowledge (Hüllermeier & Waegeman, 2021).

The TU is the entropy of the mean predictive distribution

$$\text{TU}(\mathbf{x}) = \mathbb{H}\left[\frac{1}{K}\sum_{k=1}^{K} p\Big(y|\mathbf{x}, \boldsymbol{\theta}^{(k)}\Big)\right], \tag{1}$$

where $\boldsymbol{\theta}^{(k)}$ are the model parameters of the $k$'th ensemble member or stochastic pass, and $p(y|\mathbf{x}, \boldsymbol{\theta}^{(k)})$ is its predictive distribution. The irreducible uncertainty AU is the average entropy of predictions:

$$\text{AU}(\mathbf{x}) = \frac{1}{K}\sum_{k=1}^{K} \mathbb{H}\left[p\Big(y|\mathbf{x}, \boldsymbol{\theta}^{(k)}\Big)\right]. \tag{2}$$

Finally, the reducible uncertainty EU is their difference,

$$\text{EU}(\mathbf{x}) = \text{TU}(\mathbf{x}) - \text{AU}(\mathbf{x}). \tag{3}$$

In this work we study the power-law of test predictive uncertainty against training size $N$,

$$\frac{1}{N_{\text{test}}}\sum_{n=1}^{N_{\text{test}}} \text{U}(\mathbf{x}_n), \qquad \text{U} \in \{\text{TU}, \text{AU}, \text{EU}\}. \tag{4}$$

While popular, these entropy-based metrics have limitations: (i) The standard decomposition of TU assumes additive separation of AU and EU, which Wimmer et al. (2023) argues not to hold in complex deep models, and the difficulty of disentangling them hinders their interpretability (de Jong et al., 2025); (ii) Recent works highlight EU collapse of large ensembles, leading to overly confident predictions (Kirsch, 2024; Fellaji & Pennerath, 2024).

Despite these criticisms, these metrics remain useful and tractable in practice, especially when paired with typical UQ methods. Their consistency across tasks such as active learning, out-of-distribution detection and model calibration makes them valuable diagnostic tools.

## 2.2 SCALING LAWS AND GENERALIZATION

Empirical studies in deep learning have demonstrated that performance metrics scale predictably in model size, dataset size and compute. Initial findings by Hestness et al. (2017) and Kaplan et al. (2020) showed that test loss typically decreases following a power-law of the form

$$\mathcal{L}(x) = \mathcal{L}_{\infty} + \left(\frac{x_0}{x}\right)^{\gamma}, \tag{5}$$

where $x$ is the resource under analysis (e.g., dataset size $N$, model size $P$, or compute budget $C$), $\mathcal{L}_{\infty}$ is the irreducible loss, and $\gamma$ is modality- and task-specific ($x_0$ is a reference constant). These relationships hold consistently across model architectures, tasks, and modalities, as empirically shown by Henighan et al. (2020), who demonstrated similar loss scaling for language, image, video and multimodal domains. The scaling exponents differ across domains, but the functional form of the scaling law remains stable, indicating a general underlying behavior.

Theoretical explanations of scaling behaviors are only recently emerging: for example Sharma & Kaplan (2022); Bahri et al. (2024) linked $\gamma$ to the intrinsic dimension and spectral properties of the data, showing how the data geometry drives the reducible portion of the loss. However, despite these empirical and theoretical advances, a comprehensive understanding of how EU and AU scale with model and data size remain an open question. Indeed, a Bayesian perspective on uncertainty scaling laws is currently missing: it is unclear what kind of power laws uncertainty exhibits, if any. Understanding uncertainty scaling could significantly enhance the development and comprehension of Bayesian deep learning.

## 3 METHODS

In this section, we describe the approximate Bayesian inference and ensemble methods used in our experiments. We hypothesize that if uncertainty scaling laws exist, they should emerge regardless of the UQ method.

**MC Dropout.**   MC Dropout is a simple and common inference method, with connections to VI (Gal & Ghahramani, 2016). During training, standard dropout is applied, while at test time dropout masks are resampled to produce stochastic forward passes, yielding ensemble predictions. Due to its simplicity and universality, MC Dropout is a good UQ baseline.

**Gaussian Approximations.**   Gaussian are a common family of posterior approximations, encompassing both Laplace methods (Ritter et al., 2018) and variational inference (VI) (Graves, 2011). While Laplace relies on the local curvature of the log-posterior to define the Gaussian covariance, VI instead directly optimizes an approximate distribution to match the true posterior in the Kullback-Leibler divergence (KL) sense. Building on this line of work, we also include experiments with the Improved Variational Online Newton (IVON) optimizer (Shen et al., 2024), which unifies natural-gradient VI (Khan & Rue, 2023) with an online Newton method to achieve efficient large-scale Bayesian deep learning.

**Markov Chain Monte Carlo.**   MCMC is the classic method of obtaining samples from the posterior over model parameters (Neal, 1996; MacKay, 1995). Gradient-based MCMC, such as Hamiltonian Monte Carlo (HMC), are some of the most effective samplers. Mini-batch-based SGHMC (Chen et al., 2014) and Langevin dynamics (Welling & Teh, 2011) have been successfully proposed to sample from the posterior over parameters of deep neural networks of moderate size (Tran et al., 2022; Izmailov et al., 2021). We consider parameter sampling with both global parameter priors and Gaussian process functional priors (Tran et al., 2022).

**Deep Ensembles.**   Deep ensembles (Lakshminarayanan et al., 2017) are a popular technique to obtain predictive uncertainties, despite lacking a full Bayesian interpretation. Multiple networks are trained independently from different seeds and uncertainty is estimated from the ensemble of predictions. Deep ensembles tend to provide better uncertainty estimates and are more robust to model misspecification than MC Dropout. We train ensembles of size $M \in \{5, 10\}$.

**Partially stochastic networks.**   We consider partially stochastic networks where only few layers are inferred, while rest are optimized in a standard way (Sharma et al., 2023).

## 4   EXPERIMENTS

In this section, we provide a detailed analysis of our experimental results. We first highlight the most significant findings in the vision domain, focusing on the scaling behavior of uncertainty both in-distribution and out-of-distribution (OOD). We then report an additional experiment in the text classification setting using a GPT-2 language model. Our study covers a wide range of configurations across architectures, datasets, and UQ setups, providing a comprehensive view of scaling trends. For completeness, we also include in Appendix B a toy experiment with a Bayesian multilayer perceptron (MLP).

### 4.1   IMAGE CLASSIFICATION

We use common image classification architectures of ResNet (He et al., 2016), WideResNet (Zagoruyko & Komodakis, 2016) and Vision Transformer (ViT) (Dosovitskiy et al., 2021). We report CIFAR-10 and ImageNet32 results in the main paper, and results on CIFAR-100 in Appendix B.

#### 4.1.1   RESNET AND WIDERESNET ARCHITECTURES

If not specified, we train these models for 400 epochs with Stochastic Gradient Descent (SGD) optimizer (momentum 0.9 and weight decay $5 \times 10^{-4}$); in some experiments we adopt a cosine annealing scheduler on the learning rate and in others we do not. See Appendix C for more experimental details.

In ResNets, we add dropout layers after the convolutional blocks and before the fully connected layers following Kim et al. (2023). For WideResNets, we adopt the official implementation by Zagoruyko & Komodakis (2016), where dropout is applied between the two convolutional layers within each

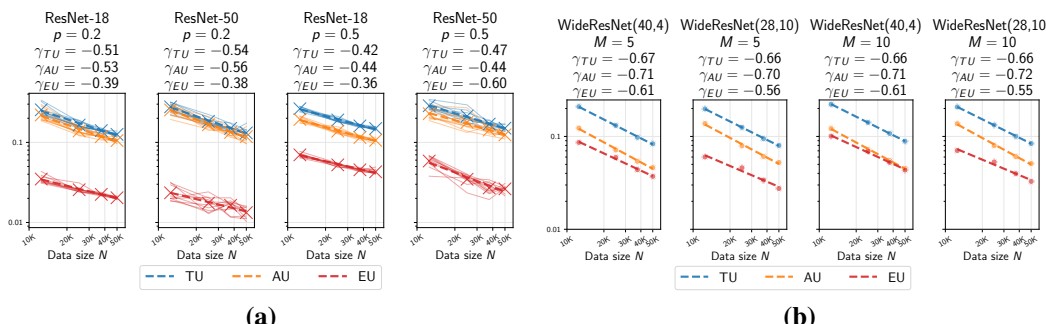

**(a)**          **(b)**

Figure 2: **ResNet and WideResNet (w,d) uncertainty scaling on `CIFAR-10` dataset**: In **(a)** we use MC Dropout with fixed dropout rate $p = 0.2$ and $p = 0.5$. In **(b)** we use Deep ensembles (with $M = 5$ and $M = 10$ ensemble members). We consider $25\%$, $50\%$, $75\%$ and $100\%$ subsets of the training data. In **(a)**, each point $\times$ corresponds to the average over 10 independent folds (varying both data subsampling and model initialization), whereas in **(b)**, the results for WideResNets are reported from a single fold. Dashed lines represent linear regressions fitted to the mean uncertainty metrics (AU, EU, TU) on a fixed test set (see Section 2.1), following a power-law decay of the form $1/N^{\gamma_{TU}}$, $1/N^{\gamma_{AU}}$, and $1/N^{\gamma_{EU}}$. Both axes are shown on a logarithmic scale.

residual unit. We experiment with various dropout rates finding more expressivity in the scaling laws obtained with $p = 0.5$, especially in ResNets architectures (Gal et al., 2017). Some results are reported in Fig. 1, Fig. 2 and Fig. 7.

We also study MC Dropout combined with SAM (Foret et al., 2020) to assess how two generalization methods interact in terms of uncertainty. We hypothesize that the increasing EU in Fig. 3 arises because SAM, by avoiding sharp minima, is forced into flatter basins as $N \to \infty$: by selecting flatter basins under increased curvature, the optimizer may avoid regions of low epistemic variance and instead favor broader regions spanning more diverse functions. This aligns with recent findings that SAM can enhance ensemble diversity by trading off sharpness minimization with data-driven de-correlation (Lu et al., 2024), while MC Dropout further promotes functional diversity via implicit bagging.

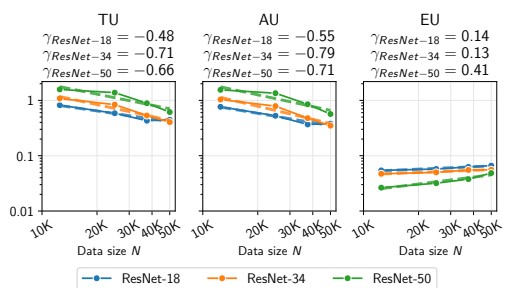

Figure 3: **Impact of SGD+SAM on uncertainty scaling**: ResNets on CIFAR-10 dataset - MC Dropout ($p = 0.5$). SAM biases solutions towards flatter minima and the combination with MC Dropout preserves functional diversity as data size increases.

Across our experiments, we observe that the EU is typically smaller than the AU. Regarding the decreasing trend of AU that we observe across various experiments, Wimmer et al. (2023) also report that AU can decrease under limited data, making estimates potentially unreliable in such regimes. Prior work (de Jong et al., 2025; Mucsányi et al., 2024) further shows that AU and EU often correlate in image classification, suggesting entanglement in practice.

Finally, we report results training on `CIFAR-10` and testing on `CIFAR-10-C` (corrupted), which allows us to explore how uncertainty behaves OOD; results in Fig. 5.

In Fig. 2, we report scaling laws for Deep Ensembles (Lakshminarayanan et al., 2017), using $M = 5$ and $M = 10$ independently trained models. Even for this UQ approach, we observe the emergence of power-law scalings of uncertainties with respect to the number of data.

We also observe a similar behavior for MCMC. In these experiments, we choose a weakly informative prior over all parameters with zero mean and standard deviation of 10. In Appendix B, we also report results for MCMC after optimizing the priors according to the approach in Tran et al. (2022), where

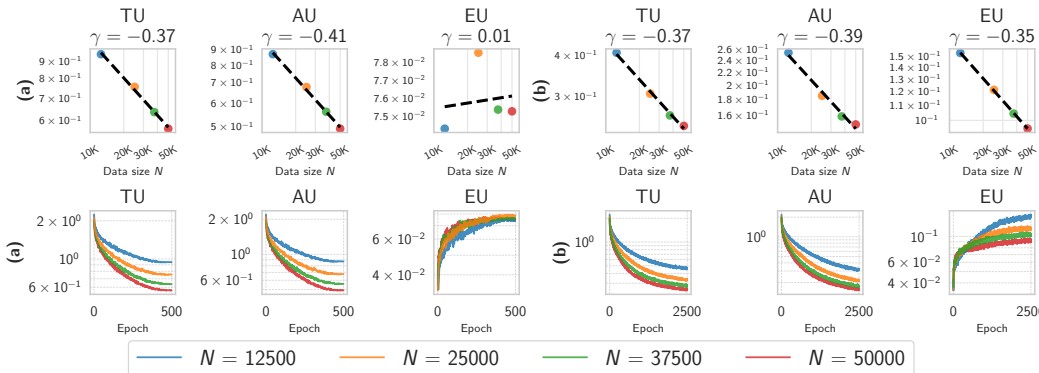

Figure 4: **ViT-small uncertainty training dynamics on `CIFAR-10` dataset:** In **(a)** we use MC Dropout (fixed $p = 0.5$ both in the embeddings and in the transformer blocks) and we train the model for 500 epochs with Adam optimizer (Kingma & Ba, 2017) and cosine annealing. In **(b)** we train the same model for 2500 epochs and fixed learning rate $10^{-4}$. The training dynamics show that the the shape/speed of convergence of of TU, AU, EU strongly depends on the optimization trajectories underneath.

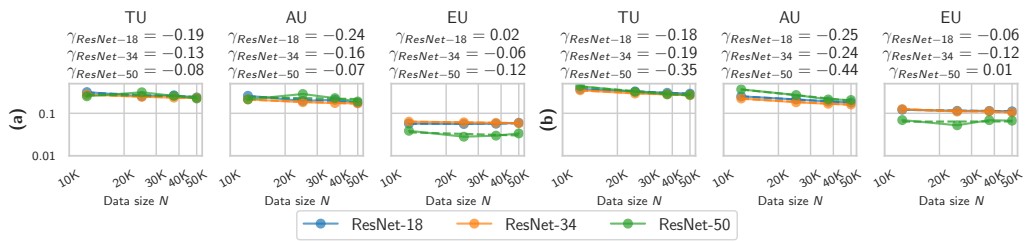

Figure 5: **ResNets on `CIFAR-10-C` dataset:** MC Dropout with $p = 0.2$ in **(a)** and $p = 0.5$ in **(b)**. For models trained on incrementally larger training subsets of `CIFAR-10`, we report the predictive uncertainties when testing on the (whole) `CIFAR10-C` dataset, averaged over all corruption levels (1-5) and corruption types considered. OOD, we expect to observe larger uncertainties - EU, for instance, should decay gradually as the data space becomes increasingly populated with additional samples within the same domain (in-fill).

we target a Gaussian process (GP) with isotropic covariance with log-length-scale of $\frac{1}{2} \log D$ and and log-marginal variance of 2. In Fig. 6, MCMC (1) and MCMC (2) refer to the approach of treating the first and first+second layers of the model in a stochastic fashion, while the other parameters are kept fixed to a pretrained solution (Sharma et al., 2023).

**Uncertainty scaling with model parameters** We investigate uncertainty scaling with model size in a preliminary experiment using MC Dropout and the IVON optimizer. IVON shows the expected increase in EU with larger models, whereas MC Dropout does not—likely reflecting limitations of the inference scheme. Parameter permutation symmetries (e.g., swapping hidden units or attention heads) generate exponentially many modes that yield the same function. Recent works (Rossi et al., 2023; Laurent et al., 2023; Gelberg et al., 2024) show these modes are connected by low-loss paths and are not functionally distinct. Thus, increasing model capacity adds redundancy rather than distinct hypotheses, explaining the weak or flat EU dependence observed in Fig. 7 **(a)**.

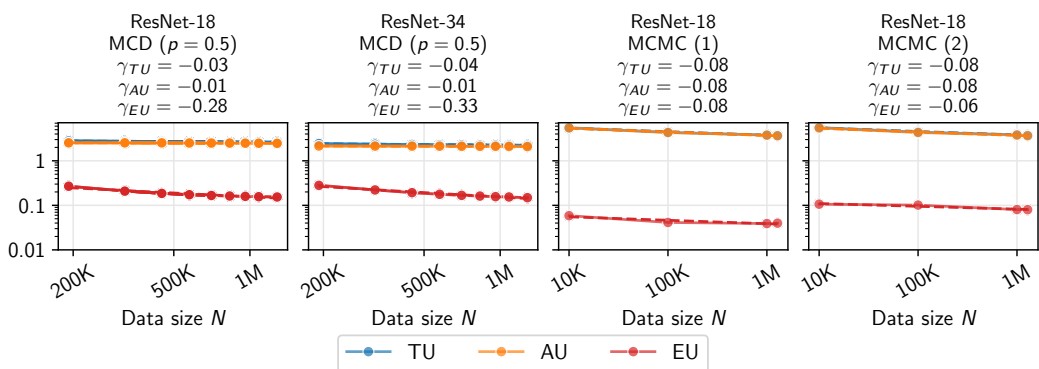

Figure 6: **Uncertainty estimates of ResNets on `ImageNet-32` dataset**: The first two subplots report MC Dropout ($p = 0.5$) uncertainties for ResNet-18 and ResNet-34, trained with fixed learning rate $10^{-3}$ on 9 increasing subsets of the training data. We also report uncertainties obtained through MCMC considering only the first layer stochastic (MCMC (1)) and only the first two layers stochastic (MCMC (2)) for ResNet-18 trained on 4 increasing subsets of the training data.

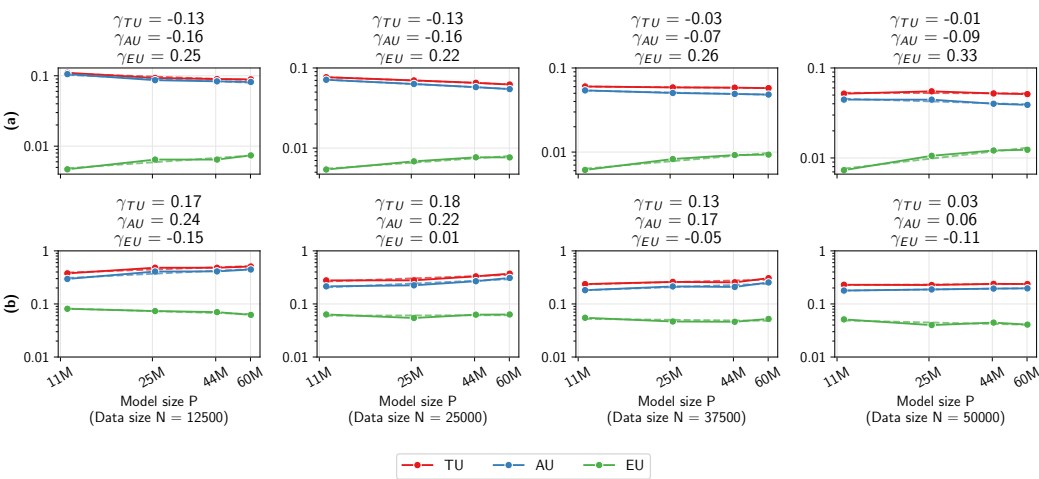

Figure 7: **Uncertainty scaling with model parameters on `CIFAR-10`:** We evaluate a wide spectrum of ResNet architectures on `CIFAR-10` dataset. The corresponding parameter counts are: ResNet-18 ($11.7M$), ResNet-50 ($25.6M$), ResNet-101 ($44.5M$), and ResNet-152 ($60.2M$). In **(a)**, we train each model with IVON optimizer for 200 epochs following the setup of Shen et al. (2024) - details in Appendix C. In **(b)**, we compare against MC Dropout ($p = 0.5$), training for $400$ epochs with SGD and fixed learning rate $10^{-3}$.

#### 4.1.2 VISION TRANSFORMER

We explore uncertainty scaling trends using Vision Transformer (ViT) architectures (Dosovitskiy et al., 2021). We conduct a study to highlight how different experimental settings lead to completely different uncertainty behaviors (Fig. 4). It is interesting to observe the effect of annealing the learning rate with a cosine scheduler when training on `CIFAR-10`, suggesting that early-phase uncertainty dynamics in transformers can be sensitive to optimization strategies. Results on `ImageNet32` are reported in Fig. 9.

## 4.2 TEXT CLASSIFICATION

We firstly investigate the language modality by assessing uncertainty estimates on the pre-trained Phi-2 model (Abdin et al., 2024), applying a Laplace approximation to the posterior over the LoRA parameters (Yang et al., 2024). We fine-tuned the model on `qqp` and `ARC` datasets observing that the uncertainties remain flat for every data subset used for fine tuning (see Fig. 15). This saturation effect is likely due to the massive amount of data used for pre-training, limiting the expressiveness of the model uncertainty on comparatively much smaller data for fine-tuning.

### 4.2.1 GPT-2 ON ALGORITHMIC DATASET

We train a GPT-2 model to solve modular arithmetic problems from a synthetic dataset, following Power et al. (2022). The model learns to predict the token after the (=) sign. We only experiment with MC Dropout with rate $p = 0.1$. Interestingly, more evident uncertainty scaling patterns only emerge after extensive training, suggesting potential links between grokking dynamics (Belkin et al., 2019) and uncertainty behavior. In Fig. 8 we show predictive uncertainties when training the model on increasing percentages (from 5% to 90%) of the `MODULO 97` algorithmic dataset. The slight uncertainty increase in the last $N$ subset of Fig. 8 likely stems from convergence dynamics and limitations of MC Dropout.

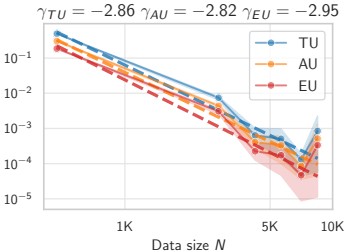

Figure 8: **Algorithmic dataset**: GPT-2 model on the Algorithmic dataset to learn the modulo addition operation. Results obtained by training for 10 000 epochs and averaged over 3 seeds.

## 5 THEORETICAL CONNECTIONS

The theory of identifiable models has extensively analyzed asymptotic behaviors, such as posterior contraction and convergence of test loss and generalization error. In this section, we recall how predictive uncertainty contracts with increasing data in Bayesian linear regression, highlight its connection to Watanabe's Generalization Error, and outline links to SLT.

### 5.1 TU SCALING IN IDENTIFIABLE PARAMETRIC MODELS

We consider Bayesian linear regression $y = \boldsymbol{\theta}^\top \boldsymbol{\phi}(\mathbf{x}) + \epsilon$, where $\boldsymbol{\theta} \in \mathbb{R}^P$ are parameters of interest, and $\boldsymbol{\phi}(\mathbf{x}) = [\phi_1(\mathbf{x}), \phi_2(\mathbf{x}), \dots, \phi_P(\mathbf{x})]^\top$ are basis functions, and assume zero-mean noise $\epsilon \sim \mathcal{N}(0, \sigma^2)$. We define the likelihood over $N$ iid observations $\{\mathbf{X}, \mathbf{y}\} = \{(\mathbf{x}_i, y_i)\}_{i=1}^N$,

$$p(\mathbf{y}|\mathbf{X}, \boldsymbol{\theta}, \sigma^2) = \prod_{n=1}^{N} \mathcal{N}\big(y_n | \boldsymbol{\phi}(\mathbf{x}_n)^\top \boldsymbol{\theta}, \sigma^2\big). \tag{6}$$

By assuming a conjugate Gaussian prior $p(\boldsymbol{\theta}) = \mathcal{N}(\boldsymbol{\theta}|\mathbf{m}_0, \mathbf{S}_0)$ the posterior is also Gaussian with mean $\mathbf{m}_N = \mathbf{S}_N \left( \mathbf{S}_0^{-1} \mathbf{m}_0 + \sigma^{-2} \boldsymbol{\Phi}^\top \mathbf{y} \right)$ and covariance $\mathbf{S}_N = \left( \sigma^{-2} \boldsymbol{\Phi}^\top \boldsymbol{\Phi} + \mathbf{S}_0^{-1} \right)^{-1}$. The predictive posterior for a new test points $(\mathbf{x}_*, y_*)$ is also Gaussian, $p(y_*|\mathbf{x}_*, \mathbf{y}, \mathbf{X}) = \mathcal{N}\left( y_* | \mathbf{m}_N^\top \boldsymbol{\phi}(\mathbf{x}_*), \sigma_N^2(\mathbf{x}_*) \right)$ with $\sigma_N^2(\mathbf{x}_*) = \sigma^2 + \boldsymbol{\phi}(x_*)^\top \mathbf{S}_N \boldsymbol{\phi}(x_*)$.

The predictive variance $\text{Var}\left[ y_* \mid \mathbf{x}_*, \mathbf{y}, \mathbf{X} \right]$ decomposes into the uncertainty of data $\sigma^2$ (AU), and the uncertainty of parameters $\boldsymbol{\phi}(\mathbf{x}_*)^\top \mathbf{S}_N \boldsymbol{\phi}(\mathbf{x}_*)$ (EU). It can be shown that $\sigma_{N+1}^2(\mathbf{x}_*) \leq \sigma_N^2(\mathbf{x}_*)$ – the posterior distribution becomes narrower as additional data points are observed (Qazaz et al., 1997). Moreover, in the limit of $N \to \infty$ we get $\sigma_N^2(\mathbf{x}) \to \sigma^2$: the predictive uncertainty converges to its irreducible AU component.

## 5.2 SINGULAR LEARNING THEORY

Our speculative theoretical link is SLT, which provides insights into the learning dynamics of highly overparameterized models. The Fisher information of deep neural networks is often singular at certain parameter values; despite forming a measure-zero subset, these singularities significantly influence learning. SLT proves that the asymptotic properties of learning are shaped by the geometry near such degenerate points (Watanabe, 2009; 2018). Here we limit ourself to connect this to the TU in Bayesian linear regression. Using Watanabe's notation

$$F_N = -\log p(\mathbf{y}_N|\mathbf{X}_N) = -\log \int p(\mathbf{y}_N|\mathbf{X}_N, \boldsymbol{\theta})p(\boldsymbol{\theta})d\boldsymbol{\theta} \tag{7}$$

is the Negative Marginal Log Likelihood (NMLL) of the data $\mathbf{X}_N$ of size $N$. The generalization error $G_N = \mathbb{E}_{p(y_{N+1}|\mathbf{x}_{N+1}, \boldsymbol{\theta}_{\text{true}})}[F_{N+1}] - F_N$ measures the expected increase in marginal likelihood when training with one additional data point (see Appendix A.2) and it can be rewritten as:

$$G_N = -\mathbb{E}_{\underbrace{p(y_{N+1}|\mathbf{x}_{N+1}, \boldsymbol{\theta}_{\text{true}})}_{p(y)}}\Big[ \log \underbrace{p\big(y_{N+1}|\mathbf{x}_{N+1}, \mathbf{X}_N, \mathbf{y}_N\big)}_{q_N(y)} \Big], \tag{8}$$

which reduces to the log posterior predictive distribution for the $(N+1)$'th datapoint given the first $N$ datapoints, under the true process $y_{N+1} \sim \mathcal{N}\Big( y_{N+1}|\boldsymbol{\theta}_{\text{true}}^{\top}\mathbf{x}_{N+1}, \sigma_{\text{true}}^2 \Big)$. By denoting the true process $p(y)$ and the posterior predictive $q_N(y)$, we can manipulate the expression in Eq. (8) to obtain:

$$G_N = \underbrace{-\frac{1}{2}\log\left(2\pi e\sigma_{\text{true}}^2\right)}_{\text{aleatoric uncertainty}} - \underbrace{\text{KL}\big[p(y)||q_N(y)\big]}_{\text{epistemic uncertainty}}, \tag{9}$$

where $\text{KL}[p(y)||q_N(y)]$ is the KL divergence between the true and the posterior predictive distribution, quantifying the EU arising from limited knowledge of the model parameters. By looking at the asymptotic expression of TU as $N \to \infty$ and by taking, without loss of generality, identity basis functions, we get:

$$\text{TU}(\mathbf{x}_{N+1}) = \mathbb{H}[q_N(y)] = \frac{1}{2}\log(2\pi e\sigma_{\text{true}}^2) + \frac{\mathbf{x}_{N+1}^{\top}\boldsymbol{\Sigma}_{X_{N+1}}^{-1}\mathbf{x}_{N+1}}{2(N+1)} + O\left(\frac{1}{(N+1)^2}\right). \tag{10}$$

As more data is collected, the predictive posterior $q_N(y)$ converges to the true predictive distribution $p(y)$. Consequently, the generalization error $G_N$ asymptotically approaches the irreducible AU. Similarly, the TU converges to the aleatoric component as the epistemic part vanishes. In Watanabe (1999) it's proved an asymptotic expansion of $G_N$ which is proportional to the effective dimensionality of the model (Appendix A.2.1).

Recent advances on the effective dimensionality of deep models (Lau et al., 2024; Chen et al., 2024) appear especially promising, and we intend to investigate such formal connections in future work. We further hypothesize that concepts from Statistical Physics may help explain the scaling behaviors we observed.

## 6 CONCLUSIONS

Inspired by recent work on scaling laws in deep learning, we investigated whether similar patterns hold for predictive uncertainties. Across vision and language tasks, we find scaling behaviors robust to architecture, posterior approximation, and hyper-parameters. Building on SLT, we provide theoretical insights connecting generalization error and predictive uncertainty, showing why information-theoretic measures scale with dataset size. However, deriving exact power-law coefficients is difficult, as they vary unpredictably with design choices. Our results suggest practical strategies for extrapolating uncertainties to large $N$, such as estimating how much data is needed for ensemble predictions to converge. They also point to applications in active learning, where one can predict the marginal uncertainty reduction from additional data to guide annotation budgets. More broadly, while scaling laws are predictive, they are not universal but depend on how the loss landscape is traversed, underscoring the need to tailor optimization strategies to each task. Future work will explore alternative theories, such as linking uncertainties in wide networks with GPs limits. We also plan to examine scaling with compute budget and model size more in depth.

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

## A APPENDIX

### A.1 TOTAL UNCERTAINTY IN BAYESIAN LINEAR REGRESSION

We consider a class of models defined as a linear combination of fixed nonlinear functions of the input variables of the form:

$$y(\mathbf{x}, \boldsymbol{\theta}) = \sum_{j=0}^{M-1} \theta_j \phi_j(\mathbf{x}) = \boldsymbol{\theta}^\top \boldsymbol{\phi}(\mathbf{x}), \tag{11}$$

where $\boldsymbol{\theta} = (\theta_0, \ldots, \theta_{M-1})^\top$ are model parameters and $\boldsymbol{\phi} = (\phi_0, \ldots, \phi_{M-1})^\top$ are known as *basis functions* which allow $y(\mathbf{x}, \boldsymbol{\theta})$ to be a non-linear function of the input $\mathbf{x}$.

We consider the target variable $y$ that is given by a deterministic function $y(\mathbf{x}, \boldsymbol{\theta})$ plus additive Gaussian noise:

$$y = \boldsymbol{\theta}^\top \boldsymbol{\phi}(\mathbf{x}) + \epsilon, \tag{12}$$

where $\boldsymbol{\theta}$ is treated as a random vector and $\epsilon \sim \mathcal{N}(0, \sigma^2)$, obtaining

$$p(y|\mathbf{x}, \boldsymbol{\theta}, \sigma^2) = \mathcal{N}(y|\boldsymbol{\phi}(\mathbf{x})^\top \boldsymbol{\theta}, \sigma^2). \tag{13}$$

We consider a supervised learning problem with $N$ input-label training pairs $\{\mathbf{X}, \mathbf{y}\} = \{(\mathbf{x}_i, y_i)\}_{i=1}^{N}$. Assuming that the data points are drawn independently from Eq. (13), the likelihood becomes:

$$p(\mathbf{y}|\mathbf{X}, \boldsymbol{\theta}, \sigma^2) = \prod_{n=1}^{N} \mathcal{N}(y_n|\boldsymbol{\phi}(\mathbf{x}_n)^\top \boldsymbol{\theta}, \sigma^2). \tag{14}$$

According to the transformation of the features introduced by the set of basis functions $\boldsymbol{\phi}$, we define the design matrix $\boldsymbol{\Phi} \in \mathbb{R}^{N \times M}$ with entries $\boldsymbol{\Phi}_{nj} = \phi_j(\mathbf{x}_n)$. We assume a conjugate prior over $\boldsymbol{\theta}$:

$$p(\boldsymbol{\theta}) = \mathcal{N}(\boldsymbol{\theta}|\boldsymbol{\mu}_0, \boldsymbol{\Sigma}_0). \tag{15}$$

From Bayes' theorem, the posterior can be derived in closed form leading to the following result:

$$p(\boldsymbol{\theta}|\mathbf{y}, \mathbf{X}) = \mathcal{N}(\boldsymbol{\theta}|\boldsymbol{\mu}_N, \boldsymbol{\Sigma}_N), \tag{16}$$

$$\boldsymbol{\mu}_N = \boldsymbol{\Sigma}_N \left( \boldsymbol{\Sigma}_0^{-1} \boldsymbol{\mu}_0 + \sigma^{-2} \boldsymbol{\Phi}^\top \mathbf{y} \right), \tag{17}$$

$$\boldsymbol{\Sigma}_N = \left( \sigma^{-2} \boldsymbol{\Phi}^\top \boldsymbol{\Phi} + \boldsymbol{\Sigma}_0^{-1} \right)^{-1}. \tag{18}$$

We are interested in the predictive uncertainty for a new test point $(\mathbf{x}_*, y_*)$ so we inspect the predictive posterior and its properties:

$$p(y_*|\mathbf{x}_*, \mathbf{y}, \mathbf{X}) = \int p(y_*|\mathbf{x}_*, \boldsymbol{\theta}) p(\boldsymbol{\theta}|\mathbf{y}, \mathbf{X}) d\boldsymbol{\theta}$$
$$= \mathcal{N}(y_*|\boldsymbol{\mu}_N^\top \boldsymbol{\phi}(\mathbf{x}_*), \underbrace{\sigma^2 + \boldsymbol{\phi}(\mathbf{x}_*)^\top \boldsymbol{\Sigma}_N \boldsymbol{\phi}(\mathbf{x}_*)}_{\mathbb{V}[y_*|\mathbf{x}_*, \mathbf{y}]}), \tag{19}$$

where $\boldsymbol{\mu}_N$ and $\boldsymbol{\Sigma}_N$ are the posterior mean and covariance matrix. From the predictive variance $\mathbb{V}[y_*|\mathbf{x}_*, \mathbf{y}]$ we identify two components: (i) $\sigma^2$ which represents the uncertainty related to the data noise (AU) and (ii) $\boldsymbol{\phi}(\mathbf{x}_*)^\top \boldsymbol{\Sigma}_N \boldsymbol{\phi}(\mathbf{x}_*)$ which is the uncertainty associated with model parameters

(EU). As we discuss in Section 5.1, by defining $\mathbb{V}[y_* | \mathbf{x}_*, \mathbf{y}] := \sigma_N^2(\mathbf{x})$ as the predictive uncertainty for a new test point $\mathbf{x}$ when training on $N$ input points, we have that $\sigma_{N+1}^2(\mathbf{x}) \leq \sigma_N^2(\mathbf{x})$, meaning that the posterior distribution becomes narrower as additional data points are observed (Qazaz et al., 1997). Moreover, $\lim_{N \to \infty} \sigma_N^2(\mathbf{x}) = \sigma^2$; that is, in the infinite data regime, the predictive uncertainty converges to its irreducible component, AU.

### A.1.1 FROM POSTERIOR PREDICTIVE VARIANCE TO TOTAL UNCERTAINTY

We now derive an expression for the (total) predictive uncertainty associated with a new data point $(\mathbf{x}, y)$, defined as the entropy of the predictive posterior:

$$
\begin{aligned}
\mathrm{TU}(\mathbf{x}) &= \mathbb{H}\left[p(y|\mathbf{x}, \mathbf{y}, \mathbf{X})\right] \\
&= \frac{1}{2} \log\left(2\pi e \sigma_N^2(\mathbf{x})\right) \\
&= \frac{1}{2} \log\left(2\pi e \left(\sigma^2 + \phi(\mathbf{x})^\top \left(\sigma^{-2}\boldsymbol{\Phi}^\top\boldsymbol{\Phi} + \boldsymbol{\Sigma}_0^{-1}\right)^{-1} \phi(\mathbf{x})\right)\right) \\
&= \frac{1}{2} \log\left(2\pi e \sigma^2 \left(1 + \sigma^{-2}\phi(\mathbf{x})^\top \left(\sigma^{-2}\boldsymbol{\Phi}^\top\boldsymbol{\Phi} + \boldsymbol{\Sigma}_0^{-1}\right)^{-1} \phi(\mathbf{x})\right)\right) \\
&= \frac{1}{2} \log\left(2\pi e \sigma^2\right) + \frac{1}{2} \log\left(1 + \sigma^{-2}\phi(\mathbf{x})^\top \left(\sigma^{-2}\boldsymbol{\Phi}^\top\boldsymbol{\Phi} + \boldsymbol{\Sigma}_0^{-1}\right)^{-1} \phi(\mathbf{x})\right) \\
&= \underbrace{\frac{1}{2} \log\left(2\pi e \sigma^2\right)}_{\mathrm{AU}(\mathbf{x})} + \underbrace{\frac{1}{2N}\phi(\mathbf{x})^\top \boldsymbol{\Sigma}_\phi^{-1}\phi(\mathbf{x})}_{\mathrm{EU}(\mathbf{x})} + O\left(\frac{1}{N^2}\right)
\end{aligned}
\tag{20}
$$

In the previous derivation, we assume $\mathbf{X} = (\mathbf{x}_1, \ldots, \mathbf{x}_N)$ to be a collection of IID samples with $\mathbf{0}$ mean and $\boldsymbol{\Sigma}_x$ covariance matrix. The same holds for $\boldsymbol{\Phi}$ where samples $\phi(\mathbf{x})$ are drawn independently from a distribution with $\mathbf{0}$ mean and $\boldsymbol{\Sigma}_\Phi$ covariance matrix. By the Law of Large Numbers $\frac{1}{N}\boldsymbol{\Phi}^\top\boldsymbol{\Phi} \to \mathbb{E}[\phi(\mathbf{x})\phi(\mathbf{x})^\top] = \boldsymbol{\Sigma}_\Phi$ as $N \to \infty$ which implies $\boldsymbol{\Phi}^\top\boldsymbol{\Phi} \to N\boldsymbol{\Sigma}_\Phi$. In addition, for large $N$ the $\boldsymbol{\Sigma}_0^{-1}$ term vanishes. This shows that as $N \to \infty$, TU approaches AU and the EU decays with rate $\frac{1}{N}$.

### A.2 TOTAL UNCERTAINTY AND GENERALIZATION ERROR FOR LINEAR MODELS

In this Section we formally derive a connection between the Watanabe generalization error $G_N$ and the Total Uncertainty (TU) for Bayesian linear models. Using the notation from Watanabe (2009; 2018) the generalization error $G_N$ and the Free Energy $F_N$ are defined as:

$$
G_N = \mathbb{E}_{p(y_{N+1}|\mathbf{x}_{N+1}, \boldsymbol{\theta}_{\mathrm{true}})}[F_{N+1}] - F_N
\tag{21}
$$

$$
F_N = -\log p(\mathbf{y}_N|\mathbf{X}_N) = -\log \int p(\mathbf{y}_N|\mathbf{X}_N, \boldsymbol{\theta})p(\boldsymbol{\theta})d\boldsymbol{\theta}.
\tag{22}
$$

We consider the same linear model of Appendix A.1 and without loss of generality we set the basis functions to be identity functions, i.e., $\phi(\mathbf{x}) = \mathbf{x}$. We also introduce $\boldsymbol{\theta}_{\mathrm{true}}$, the set of underlying true parameters. A new data point $(\mathbf{x}_{N+1}, y_{N+1})$ is generated according to the true process:

$$
y_{N+1} \sim \mathcal{N}(y_{N+1}|\mathbf{x}_{N+1}, \boldsymbol{\theta}_{\mathrm{true}}) = \mathcal{N}(y_{N+1}|\boldsymbol{\theta}_{\mathrm{true}}^\top\mathbf{x}_{N+1}, \sigma_{\mathrm{true}}^2).
\tag{23}
$$

The free energy $F_N$ is defined as the negative log marginal likelihood:

$$
\begin{aligned}
F_N &= -\log p(\mathbf{y}_N|\mathbf{X}_N) \\
&= -\log \int p(\mathbf{y}_N|\mathbf{X}_N, \boldsymbol{\theta})p(\boldsymbol{\theta})d\boldsymbol{\theta} \\
&= -\log \int \left(\prod_{n=1}^N p(y_n|\mathbf{x}_n, \boldsymbol{\theta})\right) p(\boldsymbol{\theta})d\boldsymbol{\theta}.
\end{aligned}
\tag{24}
$$

We now relate $F_{N+1}$ and $F_N$, the free energy for $N+1$ and $N$ input points, respectively:

$$
\begin{aligned}
F_{N+1} &= -\log p(Y_{N+1}|\mathbf{X}_{N+1}) \\
&= -\log p(y_{N+1}, \mathbf{y}_N|\mathbf{x}_{N+1}, \mathbf{X}_N) \\
&= -\log p(y_{N+1}|\mathbf{x}_{N+1}, D_N)p(\mathbf{y}_N|\mathbf{X}_N) \\
&= -\log p(y_{N+1}|\mathbf{x}_{N+1}, D_N) - \log p(\mathbf{y}_N|\mathbf{X}_N) \\
&= -\log p(y_{N+1}|\mathbf{x}_{N+1}, D_N) + F_N
\end{aligned}
\tag{25}
$$

where $D_N = \{\mathbf{X}_N, \mathbf{y}_N\}$ and $p(y_{N+1}|\mathbf{x}_{N+1}, D_N)$ is the posterior predictive distribution for the $(N+1)$ data point given the first $N$ data points:

$$
\begin{aligned}
p(y_{N+1}|\mathbf{x}_{N+1}, D_N) &= \int p(y_{N+1}|\mathbf{x}_{N+1}, \boldsymbol{\theta})p(\boldsymbol{\theta}|D_N)d\boldsymbol{\theta} \\
&= \mathcal{N}(y_{N+1}|\mu_{\text{pred}}, \sigma_{\text{pred}}^2), 
\end{aligned}
\tag{26}
$$
$$
\mu_{\text{pred}} := \mathbb{E}[y_{N+1}|\mathbf{x}_{N+1}, D_N] = \boldsymbol{\mu}_N^\top \mathbf{x}_{N+1}
\tag{27}
$$
$$
\sigma_{\text{pred}}^2 := \mathbb{V}[y_{N+1}|\mathbf{x}_{N+1}, D_N] = \sigma_{\text{true}}^2 + \mathbf{x}_{N+1}^\top \boldsymbol{\Sigma}_N \mathbf{x}_{N+1}.
\tag{28}
$$

Let's simplify the notation: we rename the new data point $y_{N+1}$ simply $y$ such that $p(y) := p(y_{N+1}|\mu_{\text{true}}, \sigma_{\text{true}}^2)$ where $\mu_{\text{true}} := \boldsymbol{\theta}_{\text{true}}^\top \mathbf{x}_{N+1}$. We can now compute the generalization error:

$$
\begin{aligned}
G_N &= \mathbb{E}_{p(y)}[F_{N+1}] - F_N \\
&= \mathbb{E}_{p(y)}[F_{N+1} - F_N] \\
&= \mathbb{E}_{p(y)}[-\log p(y_{N+1}|\mathbf{x}_{N+1}, D_N)] \\
&= \mathbb{E}_{p(y)}\left[ \frac{1}{2}\log(2\pi\sigma_{\text{pred}}^2) + \frac{(y - \mu_{\text{pred}})^2}{2\sigma_{\text{pred}}^2} \right] \\
&= \frac{1}{2}\log(2\pi\sigma_{\text{pred}}^2) + \frac{\sigma_{\text{true}}^2 + (\mu_{\text{true}} - \mu_{\text{pred}})^2}{2\sigma_{\text{pred}}^2} \\
&= \underbrace{\mathbb{H}[\mathcal{N}(y|\mu_{\text{true}}, \sigma_{\text{true}}^2)]}_{\text{AU}(\mathbf{x}_{N+1})} + \underbrace{\text{KL}\left[\mathcal{N}(y|\mu_{\text{true}}, \sigma_{\text{true}}^2)||\mathcal{N}(y|\mu_{\text{pred}}, \sigma_{\text{pred}}^2)\right]}_{\text{EU}(\mathbf{x}_{N+1})}
\end{aligned}
\tag{29}
$$

where we used the fact that $\mathbb{E}_{p(y)}[(y - \mu_{\text{pred}})^2] = \mathbb{V}_{p(y)}[y] + (\mu_{\text{true}} - \mu_{\text{pred}})^2 = \sigma_{\text{true}}^2 + (\mu_{\text{true}} - \mu_{\text{pred}})^2$ and:

$$
\mathbb{H}[\mathcal{N}(y|\mu_{\text{true}}, \sigma_{\text{true}}^2)] = \frac{1}{2}\log(2\pi e\sigma_{\text{true}}^2)
\tag{30}
$$
$$
\text{KL}\left[\mathcal{N}(y|\mu_{\text{true}}, \sigma_{\text{true}}^2)||\mathcal{N}(y|\mu_{\text{pred}}, \sigma_{\text{pred}}^2)\right] = \frac{1}{2}\log\left(\frac{\sigma_{\text{pred}}^2}{\sigma_{\text{true}}^2}\right) + \frac{\sigma_{\text{true}}^2 + (\mu_{\text{true}} - \mu_{\text{pred}})^2}{2\sigma_{\text{pred}}^2} - \frac{1}{2}.
\tag{31}
$$

This is closely related to the asymptotic expression we derived for TU in Appendix A.1.1 where, if we now consider identity basis functions and a test point $\mathbf{x}_{N+1}$, it becomes:

$$
\text{TU}(\mathbf{x}_{N+1}) = \underbrace{\frac{1}{2}\log(2\pi e\sigma_{\text{true}}^2)}_{\text{AU}(\mathbf{x}_{N+1})} + \underbrace{\frac{1}{2(N+1)}\mathbf{x}_{N+1}^\top \boldsymbol{\Sigma}_X^{-1} \mathbf{x}_{N+1}}_{\text{EU}(\mathbf{x}_{N+1})} + O\left(\frac{1}{(N+1)^2}\right)
\tag{32}
$$

In terms of generalization error $G_N$, as the posterior predictive $q_N(y)$ moves closer to the true data generating distribution $p(y)$, the KL term goes to zero and $G_N$ converges to the aleatoric component $\frac{1}{2}\log(2\pi e \sigma_{\text{true}}^2)$, i.e., if $\mu_{\text{pred}} = \mu_{\text{true}}$ and $\sigma_{\text{pred}}^2 = \sigma_{\text{true}}^2$ then

$$\text{KL}\left[\mathcal{N}(y|\mu_{\text{true}}, \sigma_{\text{true}}^2)||\mathcal{N}(y|\mu_{\text{pred}}, \sigma_{\text{pred}}^2)\right] = 0.$$

### A.2.1 ASYMPTOTIC BEHAVIORS

In Watanabe (1999); Hay (2025) it's showed that the generalization error and the free energy have the following asymptotic expansions:

$$\mathbb{E}_n[G_N] = \frac{\lambda}{n} - \frac{m-1}{n\log n} + o\left(\frac{1}{n\log n}\right) \tag{33}$$

$$F_N = nS_n + \lambda \log n - (m-1)\log\log n + O_p(1) \tag{34}$$

where $\lambda$ is a positive rational number, $m$ a positive integer and $\mathbb{E}_n[\cdot]$ is the expectation operator the overall dataset. The constant $\lambda$ is called the RLCT or learning coefficient, since it is dominant in the leading terms of the equations above, which represents the $\mathbb{E}_n[G_n] - n$ and $F_n - n$ learning curves. In algebraic geometry, $m$ is called a multiplicity. In the case of regular models $\lambda = \frac{d}{2}$ and $m = 1$, where $d$ is the number of model parameters.

### A.2.2 ON CONNECTING SLT TO UNCERTAINTY SCALING IN DEEP MODELS

Our use of Bayesian linear regression is primarily to illustrate how, in a simplified setting, generalization error can be cleanly decomposed into aleatoric and epistemic components, thereby exposing uncertainty measures that are otherwise less apparent. We are aware that this only scrapes the surface of the problem, as extending these insights to deep, over-parameterized networks remains an open challenge. Within the broader perspective of SLT Watanabe (2009), neural networks form a singular and non-realizable class of models, for which theoretical results—such as asymptotic error bounds or explicit learning coefficients—are scarce. In particular, Bayes and Gibbs errors are governed by the RLCT, which quantifies effective dimensionality; yet for over-parameterized architectures, the roles of RLCT and related invariants remain poorly understood despite recent advances in local approximations, such as the Local Learning Coefficient (LLC) Lau et al. (2024).

## B ADDITIONAL RESULTS

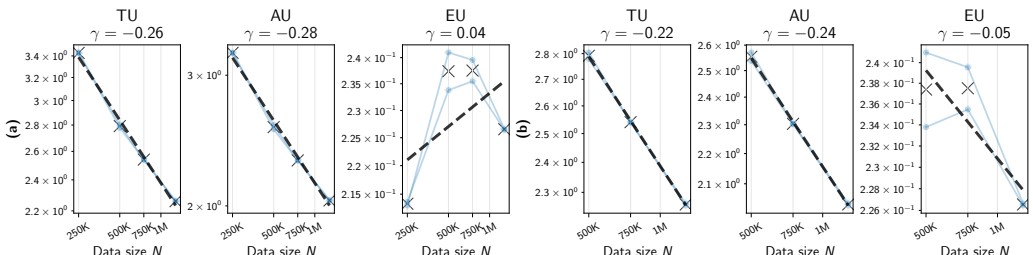

Figure 9: **Vision Transformer (ViT) on `ImageNet-32` dataset**: We use MC Dropout with fixed dropout rate $p = 0.1$. We train for 200 epochs using SGD with cosine annealing. In **(a)** we report uncertainty scaling on subsets from size $250K$ up to $1.2M$. In **(b)** we discard the first point highlighting the decreasing rate of EU for large values of $N$.

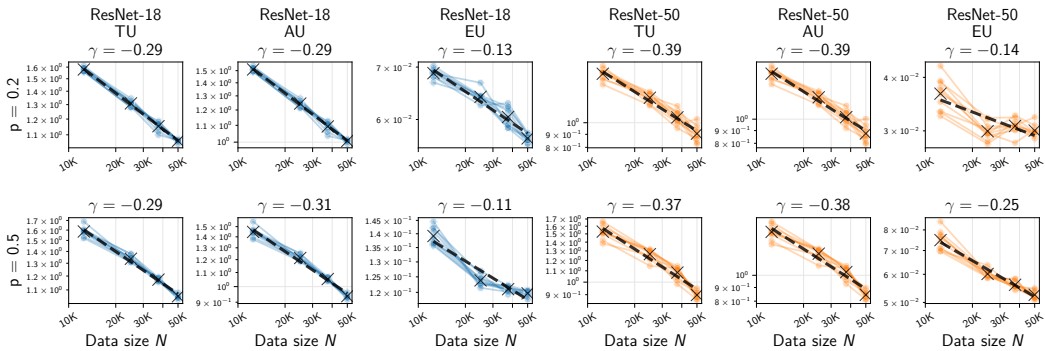

Figure 10: **ResNets on `CIFAR-100` dataset**: We train the models with MC Dropout with fixed dropout rate $p = 0.2$ (*first row*) and $p = 0.5$ (*second row*). We consider $25\%$, $50\%$, $75\%$ and $100\%$ subsets of the training data. We report results from 10 independent folds (varying both data subsampling and model initialization) showing the mean uncertainty for each $N$ subset. The dashed lines represent linear regressions fitted to the average (over the folds for every $N$) of each uncertainty metric. Both axes are on a logarithmic scale.

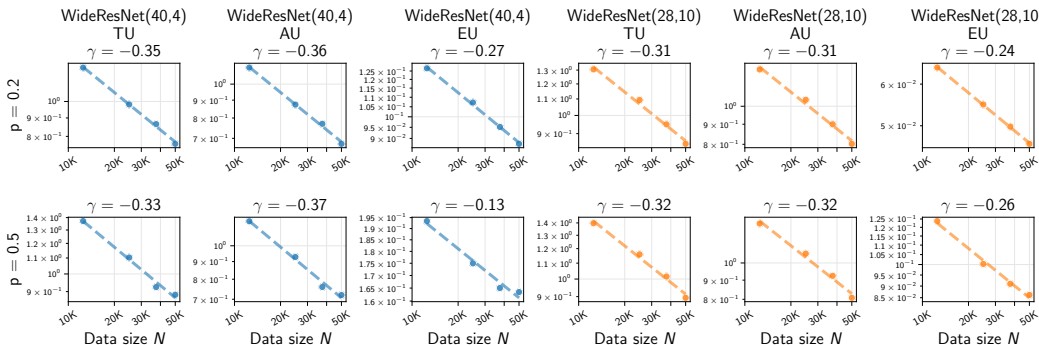

Figure 11: **WideResNets (w,d) on CIFAR-100 dataset**: We train the models with MC Dropout with fixed dropout rate $p = 0.2$ (*first row*) and $p = 0.5$ (*second row*). We consider $25\%$, $50\%$, $75\%$ and $100\%$ subsets of the training data and we show the mean uncertainty for each $N$ subset. Dashed lines represent linear regression fitted to the test mean of each uncertainty metric. Both axes are on a logarithmic scale.

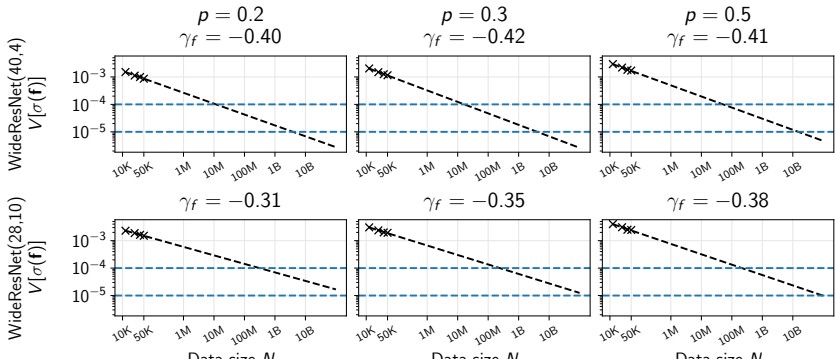

Figure 12: **WideResNets (w,d) on CIFAR-10 dataset - extrapolation**: We report uncertainty trends for different WideResNets and dropout rates. We mention in the main paper that the scaling laws we derive in this context are practically useful to extrapolate uncertainties to $N$ arbitrarily large. We report $\mathbb{V}[\sigma(\mathbf{f})]$, the variance computed over the Softmax predictions of the MC samples, averaged over all the test samples ($\times$ in the plots). The black dashed lines have slope $\gamma_f$, the blue dashed lines correspond to two thresholds of predictive uncertainty.

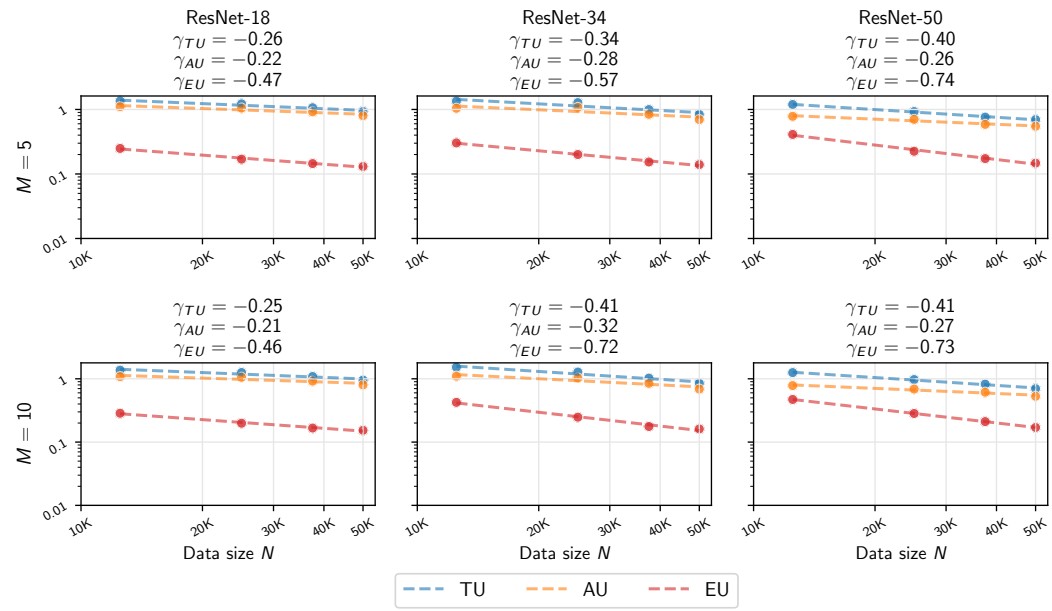

Figure 13: **ResNets on `CIFAR-100` dataset with Deep Ensembles**: We train the models with Deep Ensembles with ensemble members $M = 5$ (*first row*) and $M = 10$ (*second row*). We consider 25%, 50%, 75% and 100% subsets of the training data and we show the mean uncertainty for each $N$ subset. Dashed lines represent linear regression fitted to the test mean of each uncertainty metric. Both axes are shown on a logarithmic scale.

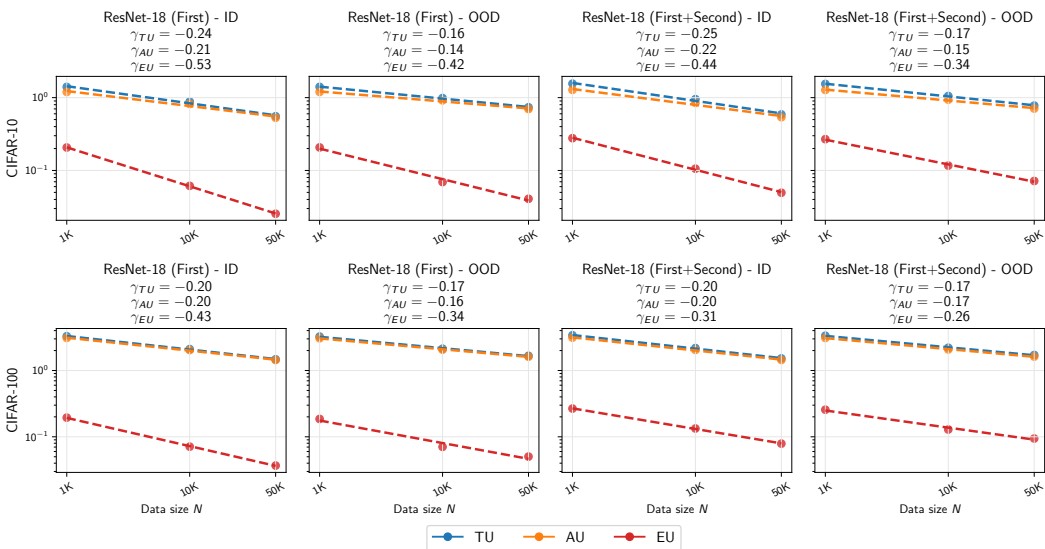

Figure 14: **ResNet-18 on `CIFAR-10` and `CIFAR-100` datasets with MCMC:** We report uncertainties obtained through MCMC considering only the first layer stochastic (First) and only the first two layers stochastic (First+Second). We show the scaling for both in-distribution (ID) and out-of-distribution (OOD) predictive uncertainties. On the first row the mean uncertainty scaling for `CIFAR-10` using `CIFAR-10-C` as OOD dataset while on the second row the mean uncertainty scaling for `CIFAR-100` using `CIFAR-100-C` as OOD dataset.

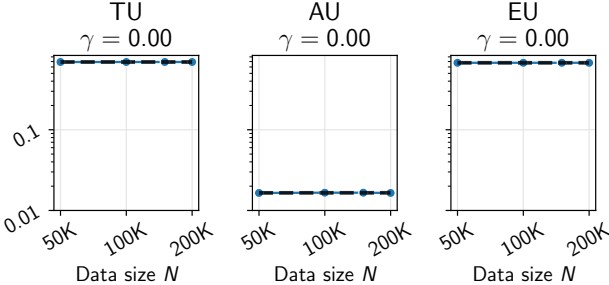

Figure 15: **Uncertainty scaling with Bayesian LoRA**: For completeness, we report the uncertainty scaling we get when fine-tuning LLM Phi on Quora Questions Pairs (`qqp`) dataset using increasing training subsets and by doing Laplace approximation over the LoRA parameters following Yang et al. (2024). These curves are completely flat, suggesting that assessing predictive uncertainty on much smaller (fine-tuning) data compared to the massive amount used for pre-training doesn't give any insight about the scaling laws of the metrics of interest.

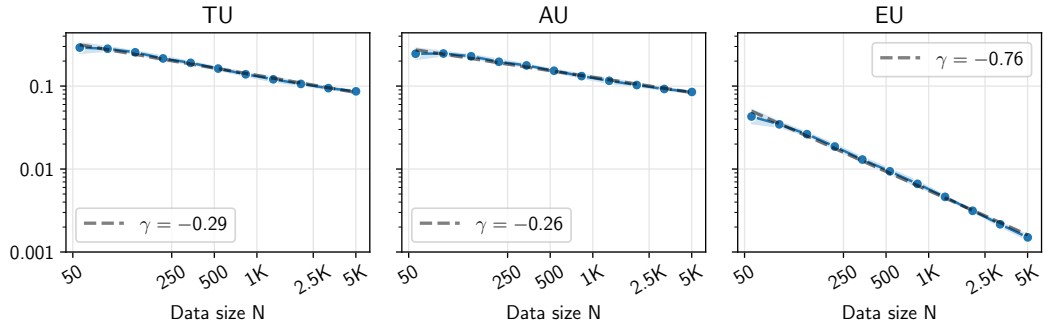

Figure 16: **Uncertainty scaling of an MLP on the two_moons dataset**: We run a toy $2D$ experiment on the two_moons dataset (10,000 samples, 5,000 train/test split). We use Bayesian logistic regression with normal priors on an MLP with two hidden layers of size 32 and $\tanh$ activation functions. Posterior inference is performed with HMC sampling (200 posterior samples after 1,000 warm-up steps). Training is carried out on 20 log-spaced values of $n_{\mathrm{train}}$ and 5 random seeds. Uncertainties are computed on the held-out 5,000 test set, and we report results only for $n_{\mathrm{train}} > 50$. The uncertainty is once again dominated by aleatoric contributions, while EU clearly decays with increasing $n$. This qualitative behavior is consistently observed across all the experiments presented in this work.

# C EXPERIMENTAL SETUP

Table 1: **Vision Transformer (ViT) architecture**: To obtain the results in Fig. 4 and Fig. 9, we use a compact ViT model with details specified in the table below. The implementation is available at https://github.com/kentaroy47/vision-transformers-cifar10.

| Parameter | Value |
| --- | --- |
| Patch size | 4 |
| Embedding dimension | 512 |
| Transformer depth | 6 |
| Number of attention heads | 8 |
| MLP hidden dimension | 256 |
| Dropout (transformer) | $\{0.1, 0.5\}$ |
| Dropout (embeddings) | $\{0.1, 0.5\}$ |

Table 2: **Algorithmic dataset**: To obtain the results in Fig. 8 we use a GPT-2 model and train it for 10.000 epochs with AdamW (Loshchilov & Hutter, 2019) optimizer (learning rate $10^{-4}$) and linear scheduler with 100 warmup steps. Implementation available at https://github.com/openai/grok/tree/main/scripts.

| Parameter | Value |
| --- | --- |
| Max sequence length | 256 |
| Embedding size | 128 |
| Number of layers | 2 |
| Number of attention heads | 4 |
| Dropout (residuals) | 0.1 |
| Dropout (embeddings) | 0.1 |
| Dropout (attention) | 0.1 |

Table 3: **IVON configuration and training setup**: To obtain the results in Fig. 7 we follow the setup of Shen et al. (2024) for image classification. The optimizer is combined with a scheduler (initial linear warm-up followed by cosine annealing). In this table we report the hyper-parameters we set following the same notation of Shen et al. (2024) and of the official implementation available at https://github.com/team-approx-bayes/ivon.

| Parameter | Value |
| --- | --- |
| Total epochs | 200 |
| Warm-up epochs | 5 |
| Batch size | 50 |
| Initial learning rate | 2.5 |
| $\beta_1$ | 0.9 |
| $\beta_2$ | $1 - 5 \times 10^{-6}$ |
| Weight decay | $5 \times 10^{-5}$ |
| Hessian init | 0.05 |
| ESS $\lambda$ | 1,281,167 |
| $N$ | 1,281,167 |
| MC test samples | 10 |

## D  LARGE LANGUAGE MODELS (LLMS) USAGE

LLMs were employed for text polishing. They were not involved in designing experiments, analyzing data, or generating scientific content; their use was limited to improving clarity and readability of the manuscript.

