# OpenReview forum: "Scaling Laws for Uncertainty in Deep Learning"
_ICLR.cc/2026/Conference — Submitted to ICLR 2026_

### Official Review · Reviewer_MsDW · 2025-10-23

**Soundness:** 2
**Presentation:** 3
**Contribution:** 2
**Rating:** 2
**Confidence:** 3

**Summary:**

The paper empirically demonstrates the existence of scaling laws associated with certain measures of predictive uncertainty with respect to dataset and model size, in the case of overparameterized models.
Specifically, authors demonstrate that the popular representation of epistemic uncertainty (EU = TU - AU), which is essentially an expected pointwise excess risk, obeys a power-law scaling law.
The authors demonstrate this observation using different problems, model architectures, and different posterior approximation techniques.

**Strengths:**

1) The paper is well-written and is easy to follow.

2) The paper provides a good empirical evaluation of the existence of uncertainty scaling laws.

**Weaknesses:**

1. My primary concern is that the paper does not relate to the PAC-Bayes literature, where similar results are already presented, seemingly.

The specific measures of uncertainty considered in this paper can be obtained through statistical risk decomposition, where each of the uncertainty measures (EU, AU, or TU) represents a certain risk. Specifically, the EU will be a difference between the total risk and the Bayes risk, and is well-known in the literature under the name of excess risk. Despite several available options to approximate the excess risk, the paper considers only one of them.

This object (excess risk and the particular approximation considered in this paper), which leads to Conditional Mutual Information, has been analyzed in the recent literature on PAC-Bayes [1, 2], and several upper bounds of varying tightness are known. From these upper bounds, it is evident that the excess risk follows a power-law relationship with N. In light of these results, I do not see how to consider the paper's contribution. Can authors comment on that?

2. The paper considers only one specific type of uncertainty measures (risk-based / proper-scoring-rule-based). However, there exist many more types of them (density-based, evidential-deep-learning-based, conformal-prediction-based, credal-set-based, etc.). However, the title does not clearly indicate this distinction and appears to be too broad.


References:

[1] - Futami, F. Epistemic Uncertainty and Excess Risk in Variational Inference. In The 28th International Conference on Artificial Intelligence and Statistics.

[2] - Alquier, P. User-friendly introduction to PAC-Bayes bounds. arXiv preprint arXiv:2110.11216.

**Questions:**

1. How do these results relate to PAC-Bayes theory? In case of overlapping, how should one consider the contribution of the paper?

2. Do these scaling laws generalize to other uncertainty measures as well (not only risk-based)?

---

> ### Author Response · Authors · 2025-11-27
>
> Dear Reviewer,
>
> Thank you sincerely for your thoughtful and insightful feedback. We appreciate the time and care you invested in evaluating our manuscript.
>
> At this stage, we prefer not to provide detailed responses to the individual questions. Instead, we will take this opportunity to revise and strengthen the paper for a future resubmission. In particular, we will carefully compare our approach with the references you suggested and incorporate them appropriately into the revised manuscript.

---

### Official Review · Reviewer_L8Sm · 2025-10-30

**Soundness:** 2
**Presentation:** 2
**Contribution:** 2
**Rating:** 4
**Confidence:** 3

**Summary:**

This paper demonstrates that, beyond the “loss following a power law with scale,” does prediction uncertainty (TU/AU/EU) also exhibit a stable power-law relationship with data volume N or model scale?

**Strengths:**

Strength:
1. sounds interesting as the author claims they provide "strong evidence to dispel recurring skepticism against Bayesian approaches"
2. Authors empirically observed the "power-law" in uncertainty estimation
3. Conducted extensive experiments

**Weaknesses:**

weakness:
1. I don't like "strong evidence to dispel" as the authors concluded their claims from empirical observations, and I think that's far enough to claim "strong evidence"
2. Eq.(9) seems incorrect
3. **The so-called “theoretical connection” holds only for identifiable linear models and offers no quantitative derivation for deep networks**
4. EU doesn’t decrease/near-zero or even positive slope: In multiple plots, the fitted exponent for epistemic uncertainty is effectively zero or positive (e.g., $\gamma_{\text{EU}}=0.01, 0.04$), which contradicts the paper’s claim of a universal $1/N^{\gamma}$ decay

**Questions:**

pls check weakness
I'll raise the score if the authors can address my concerns

---

> ### Author Response · Authors · 2025-11-27
>
> Dear Reviewer,
>
> Thank you very much for your insightful and constructive feedback. We truly appreciate the time and effort you dedicated to evaluating our work.
>
> At this stage, we prefer not to provide detailed responses to the questions raised. Instead, we would like to take this opportunity to carefully incorporate your suggestions as we improve and refine the manuscript for a future resubmission.

---

### Official Review · Reviewer_Ve2w · 2025-10-30

**Soundness:** 2
**Presentation:** 1
**Contribution:** 2
**Rating:** 2
**Confidence:** 4

**Summary:**

The paper aims to systematically analyze the scaling laws of different forms of uncertainty in various deep learning tasks. These scaling laws are investigated for varying dataset and model sizes. Specifically, the paper provides an introduction to various uncertainty quantification methods. The authors then present a comprehensive evaluation of the aforementioned methods in both visual and textual domains across different model architectures and training configurations. The experiments demonstrate that the scaling behavior of average uncertainty follows power-law trends with respect to dataset and model size. Finally, the paper establishes a theoretical connection between generalization error and total uncertainty in Bayesian linear regression.

**Strengths:**

1. The paper addresses an important research direction that has been largely overlooked in recent studies.
2. The work presents a comprehensive evaluation across both visual and textual domains, focusing on multiple model sizes, architectures, and training configurations for in-domain and out-of-domain visual tasks.
3. The authors provide a clear and detailed description of the training configurations, ensuring strong reproducibility of the experiments.

**Weaknesses:**

1. The structure of the paper is weak and requires significant improvement. For example, Figure 7 is cited on page 5 but appears on page 7, while Figure 4 appears on page 5 but is cited on page 7, etc. These inconsistencies disrupt the reading flow and make it difficult to follow the paper’s narrative, forcing the reader to constantly scroll back and forth to connect the references with the corresponding figures. Moreover, in some figures, the model is fixed and the types of uncertainty are distinguished by different colors, whereas in others, the uncertainty type is fixed and the model sizes vary by color. This inconsistency further complicates the interpretation of the figures.
2. Numerous abbreviations, citations, and notations are missing or insufficiently explained in the main part of the paper. For instance, starting from line 240, the authors use the abbreviation SAM without providing a description. Section 4.2 mentions several datasets but does not cite them. Section 5 introduces numerous complex notations that are either not defined or only partially explained in the appendix, without clear cross-references.
3. The Methods section overlooks approaches that are not based on model ensembles, such as those using the model’s probabilities [1] or hidden states [2, 3].
4. The paper lacks a systematic analysis and a clear summary of the experimental results. Section 4, which presents the experiments, reads more like a list of figures with their corresponding setups rather than providing insights or interpretations of the results.
5. The increase in epistemic uncertainty in several experiments is unclear and, moreover, appears to contradict the definition and theoretical foundations of this type of uncertainty.
6. While Section 5 makes some connections with the theory, it should be more closely linked to the experimental findings. Otherwise, the necessity of this section is unclear.
7. The experimental setup for the textual tasks needs significant refinement. The selection of the considered textual tasks and models is unclear. While the experiments in the image domain focus on classification tasks, the textual domain should also include classification tasks to allow the application of the considered methods. Otherwise, a different set of methods, models, and tasks should be considered for generative language models [4].
8. The practical usefulness of the explored scaling laws is unclear. Currently, the average absolute value of the uncertainty appears to reflect the model being overconfident rather than revealing any meaningful patterns. Moreover, changes in absolute values are not indicative different models or models trained on datasets of varying sizes, and thus cannot be directly compared [5]. It is also unclear how these findings would impact downstream performance in tasks such as OOD detection or selective classification [2].
9. The language of the paper should be refined. For example, on lines 238–239, a figure is referred with using the word “some”, which makes the reference unclear. Similarly, on line 211, the paper mentions “some” experiments without sufficient specification.


[1]. Yonatan Geifman, Ran El-Yaniv. Selective Classification for Deep Neural Networks. NIPS 2017. \
[2]. Kotelevskii et al. Nonparametric Uncertainty Quantification for Single Deterministic Neural Network. NeurIPS 2022 \
[3]. Lee et al. A simple unified framework for detecting out-of-distribution samples and adversarial attacks. NeurIPS 2018 \
[4]. Vashurin et al. Benchmarking Uncertainty Quantification Methods for Large Language Models with LM-Polygraph. TACL 2025 \
[5]. Ashukha et al. Pitfalls of In-Domain Uncertainty Estimation and Ensembling in Deep Learning. ICLR 2020

**Questions:**

1. What is the impact of the scaling laws on downstream applications, including OOD detection and selective classification?

---

> ### Author Response · Authors · 2025-11-27
>
> Dear Reviewer,
>
> Thank you sincerely for your thoughtful and insightful feedback. We appreciate the time and care you invested in evaluating our manuscript.
>
> At this stage, we prefer not to provide detailed responses to the individual questions. Instead, we will take this opportunity to revise and strengthen the paper for a future resubmission. In particular, we will carefully compare our approach with the references you suggested and incorporate them appropriately into the revised manuscript.

---

### Official Review · Reviewer_RP8j · 2025-11-01

**Soundness:** 2
**Presentation:** 4
**Contribution:** 2
**Rating:** 2
**Confidence:** 3

**Summary:**

This paper studies the scaling laws of uncertainty in deep learning. Through empirical evaluations spanning different uncertainty quantification techniques, different model architectures and tasks, the authors study the uncertainty estimated follow power-law trends with respect to various dataset sizes. They also derive a connection between generalization error in singular learning theory and total uncertainty.

**Strengths:**

- The paper is well written and organized.
- The paper deals with uncertainty quantification in deep learning, which is important for safety critical applications of deep learning models.

**Weaknesses:**

- The paper studies uncertainty scaling but even using the same models and data, the uncertainty quantified using different approaches end up being different and display different characteristics. This makes me wonder how generalizable the results would be,
- The paper provides an empirical study so its contribution offers limited advancement over the existing studies. Experiments settings considered are also limited, not capturing common model and data sizes.

**Questions:**

- Figure 1 and 2. Using the same model and supposedly similar data, I find it puzzling to see different values of uncertainty and scaling characteristics using different uncertainty techniques or using the same technique with different hyperparameters. At a high level, I’d expect the epistemic uncertainty to capture the model’s lack of knowledge and aleatoric capture to capture the inherent randomness and noise in the data-generating process. In the absence of reliable characterization of these, I’m worried about the extrapolation to scaling laws.
- By definition, aleatoric uncertainty cannot be reduced by collecting more data. You mention the potential unreliability of these estimates in limited data regimes. However, we see decreasing trends for aleatoric uncertainty across all the experiments conducted in all data size regimes. How do you explain this contradiction?
- In Figure 7, we see that the model size is not affecting epistemic uncertainty. That is counterintuitive to me, could you please explain?
Could you point me to the discussion in the manuscript where the claim on lines 27-28: “Our findings show that “so much data” is typically not enough to make epistemic uncertainty negligible.” is inferred?
- Figure 8. Can the authors comment on the spike starting near the end? It is common to see 10s of millions of data points in post training of LLMs and it would be helpful to extend the data size beyond 10K to understand this trend change.

---

> ### Author Response · Authors · 2025-11-27
>
> Dear Reviewer,
>
> Thank you very much for your insightful and constructive feedback. We truly appreciate the time and effort you dedicated to evaluating our work.
>
> At this stage, we prefer not to provide detailed responses to the questions raised. Instead, we would like to take this opportunity to carefully incorporate your suggestions as we improve and refine the manuscript for a future resubmission.

---

### Meta-Review · Area_Chair_ro17 · 2026-01-17

**Summary:**

This paper investigates whether predictive uncertainty in deep learning—analogous to test-loss scaling laws—also follows power-law scaling with dataset size and model size. It argues that while identifiable parametric Bayesian models can yield classical contraction rates (e.g., O(1/N)), over-parameterized deep models may behave differently, motivating an empirical study. The submission reports empirical scaling laws for multiple uncertainty estimators across vision and language settings.

Four reviewers reached a consensus to reject this paper. AC respects reviewers' opinions.

**Reviewer Concerns:**

Rebuttal status: The authors explicitly chose not to provide detailed responses to reviewer questions, and instead stated they will use feedback to improve for a future resubmission.

All concerns remain outstanding.

**Reviewer Scores:**

will likely no change.

---

### Decision · Program_Chairs · 2026-01-26

Reject